# Detection of Hypoxia in Cancer Models: Significance, Challenges, and Advances

**DOI:** 10.3390/cells11040686

**Published:** 2022-02-16

**Authors:** Inês Godet, Steven Doctorman, Fan Wu, Daniele M. Gilkes

**Affiliations:** 1The Sidney Kimmel Comprehensive Cancer Center, Department of Oncology, The Johns Hopkins University School of Medicine, Baltimore, MD 21231, USA; ines.godet@jhu.edu; 2Department of Chemical and Biomolecular Engineering, The Johns Hopkins University, Baltimore, MD 21218, USA; sdoctor1@jhu.edu (S.D.); fwu30@jhu.edu (F.W.); 3Johns Hopkins Institute for NanoBioTechnology, The Johns Hopkins University, Baltimore, MD 21218, USA; 4Cellular and Molecular Medicine Program, The Johns Hopkins University School of Medicine, Baltimore, MD 21231, USA

**Keywords:** hypoxia, detection, cancer, HIF

## Abstract

The rapid proliferation of cancer cells combined with deficient vessels cause regions of nutrient and O_2_ deprivation in solid tumors. Some cancer cells can adapt to these extreme hypoxic conditions and persist to promote cancer progression. Intratumoral hypoxia has been consistently associated with a worse patient prognosis. In vitro, 3D models of spheroids or organoids can recapitulate spontaneous O_2_ gradients in solid tumors. Likewise, in vivo murine models of cancer reproduce the physiological levels of hypoxia that have been measured in human tumors. Given the potential clinical importance of hypoxia in cancer progression, there is an increasing need to design methods to measure O_2_ concentrations. O_2_ levels can be directly measured with needle-type probes, both optical and electrochemical. Alternatively, indirect, noninvasive approaches have been optimized, and include immunolabeling endogenous or exogenous markers. Fluorescent, phosphorescent, and luminescent reporters have also been employed experimentally to provide dynamic measurements of O_2_ in live cells or tumors. In medical imaging, modalities such as MRI and PET are often the method of choice. This review provides a comparative overview of the main methods utilized to detect hypoxia in cell culture and preclinical models of cancer.

## 1. Introduction

Regions of hypoxia arise in 90% of solid tumors as cancer cells quickly proliferate, and scarce, newly formed vasculature fails to supply sufficient oxygen [1]. In patients with breast cancer, the mean partial pressure of oxygen (PO_2_) in breast tumors ranges from 2.5 to 28 mmHg, with a median value of 10 mmHg (1% O_2_), while normal human breast tissue has a median value of 65 mmHg (8% O_2_) [2]. Several studies have demonstrated that patients with hypoxic tumors have an increased risk of metastasis and mortality [3,4,5]. Hypoxia has been reported to be an adverse prognostic indicator, independent of clinical stage, at the time of diagnosis [6]. Cancer cells can adapt and survive under oxygen deprivation, and the most well-reported mechanism involves the hypoxia-inducible factors (HIFs) [7]. In an O_2_-rich environment, prolyl hydroxylases (PHDs) hydroxylate HIF-1α and HIF-2α. The von Hippel–Lindau (VHL) E3 ubiquitin ligase ubiquitinates hydroxylated HIF-1α and HIF-2α causing its proteasomal degradation [8]. In contrast, under hypoxia, HIF-1α and HIF-2α subunits become stabilized and bind to the HIF-1β subunit [9]. Both HIF-1 and HIF-2 heterodimers recognize and bind to the 5′-ACGTG-3′ enhancer sequence, resulting in the transcriptional regulation of more than a thousand genes [10,11]. The abundance and activity of both HIF-1α and HIF-2α can also be enhanced due to post-translational modifications [12]. HIF-regulated genes have been associated with angiogenesis, apoptosis, cell proliferation, cell survival, metabolism, invasion, metastasis, altered pH, and chemoresistance [5].

Hypoxia has been detected using both direct and indirect methods in cells cultured in the laboratory and animal and human tumors. Direct O_2_ measurements have been made in solid tumors of cancer patients using needle-type O_2_ electrodes [2,13,14,15]. Indirect methods, such as the immunolabeling of HIF-1/2α or downstream HIF-targets, have been used to detect potential regions of hypoxia in fixed tissue after surgical resection or biopsy [16]. Moreover, exogenous 2-nitroimidazole probes, such as pimonidazole, can be delivered to animals or humans and incorporate into hypoxic adducts that can be immunolabeled once the tissue is harvested and fixed [17]. Immunolabeling-based methods are insightful but limited by protein turnover and fixation artifacts, and they cannot be used for real-time assessment. For preclinical studies, several groups have used DNA constructs that cause the cell to express reporters, such as fluorescent proteins or luciferase, in a HIF-dependent manner as an indirect real-time readout of hypoxia [18,19]. In preclinical and clinical approaches, intratumoral O_2_ levels are measured in real-time, albeit indirectly, using techniques such as Magnetic Resonance Imaging (MRI) and Positron Emission Tomography (PET) [5].

Over the past two decades, multiple experimental approaches have been established to investigate hypoxia. To recreate intratumoral hypoxia in a laboratory setting, one fairly standard approach is to culture cancer cells in chambers with a controlled environment containing 1% O_2_ for 24 h or 48 h [20,21]. More recently, 3D models, such as spheroids or organoids, have been used to recreate spontaneous gradients of nutrients and oxygen [22]. Furthermore, murine models are a well-established approach to assess tumor growth and metastasis and also contain gradients of O_2_ with levels approaching less than 1% in peri-necrotic regions [23,24,25]. As research into the role of hypoxia in cancer progression and metastasis has expanded, models to recapitulate this condition and tools to detect it have rapidly progressed. 

This review article highlights well-established methods and emerging technologies to detect hypoxia in vitro, in cells and 3D models; in vivo, in tumors and live tissue; and ex vivo in tumor sections. In addition, we describe and categorize the most common methods by their application.

## 2. Detection of Hypoxia In Vitro 

### 2.1. Immunolabeling of Endogenous Markers

HIF-1α immunolabeling using immunohistochemical (IHC) or immunofluorescent (IF) techniques have frequently been employed to identify hypoxic cells in 2D and 3D cell culture (1 in Figure 1). In formalin-fixed, paraffin-embedded, MCF7 cell-derived spheroids cultured under 20% O_2_, IHC staining of HIF-1α was used to identify regions of hypoxia within the spheroid core. Likewise, Ohnishi et al. reported HIF-1α staining in cells localized within the center of glioblastoma (T98G) cell-derived spheroids [26]. 

HIF-1/2α immunolabeling can be challenging due to the short half-life of HIF-1/2α proteins upon re-exposure to oxygen. Moreover, HIF-1α and HIF-2α are localized in the nucleus under hypoxic conditions, making nuclear permeabilization required for their detection. This has led to the use of HIF-transcriptional targets, which are more abundant or easier to immunolabel, as an alternative target for HIF immunolabelling (2 in Figure 1). Glucose transporter 1 (GLUT-1), monocarboxylate transporter 1 (MCT-1), and carbonic anhydrase IX (CA-IX) are well established HIF-target genes [27]. Under hypoxia, cancer cells switch to aerobic respiration due to the Warburg effect [28], and GLUT-1 is upregulated to facilitate glucose uptake and catalyze the transformation of pyruvate to lactate. In addition, the expression level of MCT-1 and CA-IX increases to mediate lactate efflux and maintain an acidic extracellular environment [29,30]. Studies have correlated the expression of HIF-1α and downstream targets using immunolabeling to validate their use as a method to detect hypoxic cells. For instance, Miranda-Gonçalves et al. cultured glioma cells under 1% O_2_ for 24 h or in a spheroid model and showed that HIF-1α staining colocalized with increased GLUT-1, MCT-1, and CA-IX staining [27]. Overall, immunolabeling of HIF-1α and its downstream targets can be used as a proxy to detect hypoxia in cells exposed to low O_2_ concentrations and to detect spontaneous O_2_ gradients that develop in 3D culture over time. The limitations are that these are indirect measurements of hypoxia, do not provide the specific O_2_ concentration of the tumor, cannot be performed in real-time, and require tissue or cell fixation. It is also important to remember that multiple mechanisms other than hypoxia can enhance HIF-1α and HIF-2α expression [31].

### 2.2. Immunolabeling of Exogenous Markers

Another well-established strategy to detect hypoxia is the use of exogenous markers. These probes are a class of 2-nitroimidazoles first reported to form hypoxic adducts in CHO cells by Varghese et al. in 1976 [32] (3 in Figure 1). Under hypoxia (PO_2_ < 10 mmHg, approximately <1.3% O_2_), nitroimidazole binds to macromolecules such as proteins, peptides, and amino acids, to form adducts with their thiol groups. After reducing its imidazole ring, this probe selectively accumulates in hypoxic cells. Subsequently, hypoxic cells can be visualized by introducing a tag incorporated via an antibody that specifically recognizes the metabolic product of nitroimidazole [17]. Derivatives of nitroimidazole, such as pimonidazole and EF-5, are well-established for immunolabeling. However, other derivatives such as CCI-103F and A2-nitroimidazole (NITP) are not used due to their poor solubility in water. Another nitroimidazole, Misonidazole, lacks an antibody with high enough affinity for detection [33,34].

Both Pimonidazole and EF-5, and the specific antibodies designed to detect them, Hypoxyprobe™ and ELK3-51, respectively, are commercially available and used for over 20 years (see Section 3.2). For example, Hypoxyprobe™ and ELK3-51 staining have been used to detect hypoxia in the center of 3D cell-derived spheroids cultured in the presence of pimonidazole or EF-5, respectively [35]. In summary, exogenous hypoxia markers are useful to detect hypoxia. Nevertheless, they require pre-exposure to a nitroimidazole derivative for several hours, followed by fixation and immunolabeling with an antibody that recognizes the nitroimidazole derivative with high specificity. Moreover, they cannot be used for real-time measurements.

### 2.3. Phosphorescent Reporters

While insightful, immunolabeling of endogenous and exogenous hypoxic markers requires a fixation step, which prevents the dynamic monitoring of hypoxia. Oxygen-dependent phosphorescence quenching is a direct and noninvasive method to measure O_2_ concentrations in live cells [36]. The phosphorescence lifetime is the average time a molecule remains in an excited state until it returns to the ground state by emitting a photon. The time is linearly dependent on the level of O_2_ in the microenvironment and can be converted to a measurement of partial pressure (PO_2_) (4 in Figure 1) [37]. Multiple metal complexes have been used as phosphorescent probes, including Ru(II), Ir(III), Pt(II), Re(II), and Os(II) complexes, as well as metalloporphyrins with Pt(II) or Pd(II). Particularly Pt(II) and Pd(II) porphyrins and Ru(II) and Ir(III) complexes are the most established probes utilized in live systems (cell culture and animal models) [38,39]. Phosphorescent probes are often modified with lipophilic polymers and peptide ligands to increase cellular uptake [40,41], biocompatibility [42], and targeted organelle specificity [43]. 

Several approaches using phosphorescent lifetime probes have been established and optimized to detect O_2_ levels in cells. For instance, Kurokawa et al. used the phosphorescent dye, PtTCPP, to detect phosphorescence lifetime in Colon26 cells exposed to decreasing O_2_ concentrations (20 to 1% O_2_) or cultured in a spheroid model, using a laser scanning confocal microscope for detection. The decrease in the phosphorescence lifetime correlated with decreasing O_2_ concentrations [44]. Raza et al. established a small-molecule platinum (II) complex probe (PtLsCl) to map the O_2_ distribution across live spheroids derived from human melanoma cell lines, validated by Hypoxyprobe™ staining [45]. 

Phosphorescence lifetime imaging (PLIM) typically requires a pulsed laser to excite the phosphorescent probe, a detection system to capture and quantify the emitted photons, such as a photomultiplier tube, and software to convert captured photons into a measurement of PO_2_. Recently, a new class of phosphorescent reporters detected using a fluorescent microscope was developed. The photophysical properties of some of these O_2_ sensors have been recently reviewed and go beyond the scope of this review [46,47]. 

At least one phosphorescent probe used to detect hypoxia has been commercialized. For example, the Image-iT™ red hypoxia reagent manufactured by Thermofisher is based on an iridium complex that can be excited with 490 nm light and emits at 610 nm wavelength [48]. This probe begins to fluoresce when O_2_ levels are less than 5%, and the signal increases with decreasing O_2_. The fluorogenic response is reversible upon reoxygenation. Ilana et al. employed the Image-iT™ red hypoxia reagent to visualize and quantify hypoxia in MCF7-derived spheroids exposed to decreasing O_2_ levels from 20% to 1% [49]. Our study used it to image O_2_ gradients in murine breast tumor organoids after 11 days in culture under normal O_2_, which correlated with our fluorescent hypoxia fate-mapping reporter system [19]. In summary, the phosphorescent probes must be added before each measurement, allowing for short-term but real-time detection of O_2_ gradients. 

### 2.4. Fluorescent Reporters

An alternative approach to dynamically monitor hypoxia is the incorporation of vectors that drive the production of fluorescent proteins exclusively under hypoxia. Multiple groups have developed functional vectors containing one or more copies of a hypoxia-responsive element (HRE) located proximal to a minimal promoter, such as mCMV [50] or minTK [19] (5 in Figure 1). Incorporating an HRE causes the HIF-complex to drive the production of a protein of interest, such as GFP [19] or RFP [51], under hypoxia. The expression of such fluorescent proteins allows for the detection of hypoxic cells. However, some fluorophores require molecular O_2_ for proper protein folding and chromophore maturation. Moreover, while HIF-1/2α degradation occurs within minutes upon reoxygenation, fluorescent reporters have a longer half-life [52,53]. To decrease the half-life of a reporter, destabilization elements like the PEST sequence motif (peptide sequence that promotes ubiquitination and protein degradation)^5^ or the oxygen-dependent degradation domain (ODD) sequence derived from HIF-1α (oxygen-dependent degradation domain) are often fused to the C- or N- terminus of the reporter to accelerate its degradation when O_2_ is present [19,53,54]. Multiple variations of these biosensors have been widely and successfully utilized to investigate hypoxia in vitro. For instance, Le et al. combined a HIF-regulated GFP-PEST reporter with a mCherry-geminin reporter to investigate the metabolic phenotypes of hypoxic and reoxygenated cells [55]. Henning et al. used a GFP hypoxia reporter to show the development of a hypoxic core in HCT116 spheroids 96 h after spheroid formation [54]. 

While these methods allow for the visualization of hypoxic cores in 3D culture, once a cell is reoxygenated, the fluorescence signal is designed to be degraded. While this is helpful for the real-time monitoring of hypoxia, tracking and isolation of reoxygenated cells is not possible. Our group developed a dual vector hypoxia fate-mapping system to track hypoxic cells even after reoxygenation to overcome this obstacle. Under hypoxia, HIF stabilization promotes the transcription of a CRE-ODD complex. CRE-ODD protein expression causes Cre-Loxp recombination resulting in the excision of any gene that is ‘floxed’. We paired the HIF-CRE-ODD construct with a construct containing a floxed DsRed gene proximal to an eGFP gene. This caused the loss of DsRed expression and permanent eGFP expression under hypoxia. We transduced both MCF7 and MDA-MB-231 breast cancer cell lines with lentiviral vectors containing the two constructs. Spheroids derived from both cell lines develop spontaneous hypoxic cores over 15 days, and GFP+ cells invaded the spheroid regions with higher O_2_ content than the core of the spheroid (1% O_2_ as measured with an optical needle probe (see Section 2.6). We also developed a transgenic mouse model based on this system. The organoids derived from the spontaneous tumors that the mice developed also showed hypoxic cores when they were cultured in 3D [19]. GFP expression correlated with O_2_ concentrations in the gel as low as 1% as measured using optical oxygen detection nanoprobes (see Section 2.5). Overall, fluorescent reporters offer a reliable but indirect method for evaluating hypoxia in vitro. Although they cannot provide a specific O_2_ measurement, they can be designed to fluoresce when cells experience specific O_2_ concentrations.

### 2.5. Nitroreductase-Sensitive Fluorescent Probes

Instead of delivering a DNA construct to drive the production of a fluorescent protein under hypoxia, some groups are testing bioreductive activators of fluorescence that trigger fluorescence in specific substrates, predominantly p-substituted aromatic nitro compounds [56]. Nitroreductases (NTRs) are enzymes expressed only under hypoxic conditions that reduce the nitro group of the substrate by converting it into an amino group, causing the substrate to fluoresce [57] (6 in Figure 1). These substrates are directly added to cell culture media for in vitro use or injected into the tail vein in murine models (see Section 3.4). Since NTR fluorescent hypoxia probes are easily incorporated in cell culture, commercial options are available. For example, BioTracker™ 520 Green Hypoxia Dye by Merck can be fluorescently imaged and begins to fluoresce at 5% O_2_ [58]. This probe has successfully been employed to detect hypoxia in cells cultured as 2D and 3D spheroids [59]. 

Overall, this approach provides an indirect fluorescent readout of hypoxia. To date, multiple NTR probes have been designed and implemented, but current efforts focus on simplifying synthesis protocols and enhancing emission wavelengths [60].

### 2.6. Noninvasive Optical Oxygen Sensors

Oxygen levels can be directly monitored during cell culture by using noninvasive O_2_ sensors that are placed in cell culture wells. Light-sensitive O_2_ sensors or dyes become excited, for instance, with blue (450–495 nm) or red light (610–630 nm), and either emit light at a measurably higher wavelength or are quenched by O_2_. The emitted light is captured, and the ratio of captured light versus the emitted light can be converted to a specific O_2_ concentration [61]. The sensors can be arranged into arrays or films inside a sticky patch or foil so that they can be easily attached to the top or bottom of a transparent tissue culture plate. For instance, Silva et al. used the planar O_2_ sensor foil SF-RPsSu4 in conjunction with a novel 3D-printed ramp to assess O_2_ levels at different heights in a single well of a 96-well plate. The authors reported that the O_2_ concentration at the top of a confluent well of A549 cells cultured for 96 h was 12.9% O_2_ versus 7.5% O_2_ at the surface of the cell layer, which was confirmed with a fiber-based needle microsensor [62] (see Section 2.6). 

Commercial options using this approach are available. For example, Lewis et al. monitored O_2_ levels non-invasively by placing commercially available sensor patches manufactured by PreSens at the bottom of a tissue culture well-containing hydrogels that were incorporated with murine sarcoma (KIA) GFP-expressing cells. Three days after seeding, the authors measured O_2_ gradients from the surface (13% O_2_) to the bottom (4% O_2_) of their hypoxic gel [63]. Our studies have utilized oxygen nanoprobes (OXNANO) manufactured by Pyroscience to detect the O_2_ concentration within Matrigel matrices containing organoids derived for mouse breast tumors. The nanoprobes were dispersed in the Matrigel and excited by a red-light source (610–630 nm) using an O_2_ meter that collects the light emitted by the nanoprobes in the near-infrared range (760–790 nm). This technology allowed us to make daily measurements of O_2_ concentrations within the Matrigel and demonstrated that O_2_ levels declined from 15% to 1% O_2_ over an 11-day time course. This correlated with larger hypoxic cores that were visualized using our fluorescent hypoxia reporter system and the Image-iT™ red hypoxia probe [19]. Noninvasive optical sensors are useful for O_2_ measurements because of their accessibility and relative accuracy.

### 2.7. Invasive Optical Oxygen Sensors

To determine the O_2_ concentration at different sites within 3D structures such as organoids or spheroids, invasive O_2_ needle probes are needed. Light-sensitive dyes can be coated onto the tip of a needle probe for this purpose (see Section 3.12) and can be used to measure O_2_ as described above (see Section 2.5) [64]. Our group utilized a retractable-needle fiber-optic microprobe (Pyroscience) to demonstrate that O_2_ levels drop from the periphery (11% O_2_) to the core (1% O_2_) of MDA-MB-231 spheroids encapsulated in a collagen gel after they had been cultured for 20 days. The measurement correlated with the activation of our fluorescent hypoxia reporter system and Image-iT™ red hypoxia staining [19]. Overall, while invasive probes accurately enable 3D measurements of hypoxia gradients, their invasive nature may disrupt the physical integrity of the system being tested [63].

## 3. Detection of Hypoxia In Vivo 

### 3.1. Immunolabeling of Endogenous Markers

Immunolabeling of HIF-1α and HIF-2α, and their downstream targets, are well-established methods used to identify regions of intratumoral hypoxia, not only in vitro as described in Section 2.1, but also in tissue sections using both IHC and IF techniques [65] (1 in Figure 2). The caveat of staining for HIFs and HIF-target gene expression is that HIFs can also be regulated by many non-canonical pathways such as nutrient deprivation, oxidative phosphorylation inhibitors, and mutations in the VHL gene [31]. 

### 3.2. Immunolabeling of Exogenous Markers

Exogenous probes, as first described in Section 2.2, can be used to detect regions of hypoxia in tissue, cells, 3D organoids, or spheroids that have been fixed. To use exogenous probes in vivo, such as pimonidazole, the probe must be injected or orally administered to the subject (animal or human). The probe causes hypoxic adducts to form that can then be detected using a probe-specific antibody by IHC or IF once the tissue has been fixed and sectioned for staining [17] (2 in Figure 2). 

Recently, probes have also been developed that do not require antibody labeling. For example, Seelam et al. used U87MG and CT-26 cells xenografted into the right shoulder of BALB/c mice to compare two 2-nitroimidazole-fluorophore-conjugated derivatives comprised of FITC and RITC fluorophores, 1-(3′,6′-Dihydroxy-3-oxo-3H-spiro[isobenzofuran-1,9′-xanthene]-5-yl)-3-(2-(2-nitro-imidazolyl)ethyl)thiourea and 2-(3,6-Bis(diethylamino)-9H-xanthan-9-yl)-5-(3-(2-(2-nitro-1H-imidazole-1-yl)ethyl)thioureido)benzoic acid, respectively, with Hypoxyprobe™ (pimonidazole-HCl). After two weeks, each probe was injected intravenously with pimonidazole into mice, and tissue specimens were collected, immunolabeled for Hypoxyprobe, and imaged. The fluorescence signal from the two nitroimidazole fluorescent derivates colocalized with pimonidazole-Hypoxyprobe™ staining. Since the conjugated-derivatives have the benefit of being directly detected via fluorescence imaging ex vivo, this avoids the extra steps of immunolabeling and may offer a promising alternative to Hypoxyprobe™ immunolabeling [66]. The fluorescence stability of the FITC and RITC conjugated probes was greater than 2 h in vitro, suggesting that these probes could potentially be used for short-term fluorescent imaging, for instance, via intravital or whole-body fluorescence imaging. 

### 3.3. Fluorescent Reporters

Cell lines that have been engineered to express a fluorescent reporter after exposure to hypoxia, as introduced in Section 2.4, can also be used to generate tumors in mice in which spontaneous hypoxic regions can be imaged (3 in Figure 2). For example, Wang et al. generated an MDA-MB-231 cell line expressing a DNA construct (GFP-5HRE-ODD-mCherry) that constitutively expressed GFP but only expressed mCherry under hypoxic conditions. Using intravital imaging, the authors could dynamically image single hypoxic cells in an orthotopic tumor on a live mouse [52]. In another study, Erapaneedi et al. investigated a novel family of fluorescent reporters, UnaG, which does not require O_2_ for chromophore maturation and emits green fluorescence when bound to its ligand, bilirubin. This heme metabolite can be supplemented in cell culture medium [67]. To incorporate O_2_-dependent regulation, the authors developed a reporter construct that contained five copies of the VEGF HRE sequence to drive the hypoxia-dependent activation of UnaG. By orthotopically transplanting Gli36 reporter cells into SCID mice, the study confirmed that UnaG expression could be imaged at single-cell resolution in the brain of live mice using a cranial window [50].

The fluorescent reporters described shut down upon re-exposure to O_2_, for example, when a hypoxic cancer cell enters the more oxygenated bloodstream [68]. To overcome this obstacle and investigate the role of intratumoral hypoxia in metastasis, our group developed a dual vector hypoxia fate-mapping system as described in Section 2.4. In this system, cells that experience hypoxia maintain permanent eGFP expression. Our studies showed a correlation of eGFP, Hypoxyprobe™, and HIF-1α expression in IF-labeled tissue sections of orthotopic tumors harvested from the mammary fat pad of mice [19,69]. Furthermore, the system allowed us to image and collect cells that experienced hypoxia in the primary tumor and metastasized to distant organs such as the lung and liver [19,70]. Another fate-mapping approach by Vermeer et al. used a HIF-1α-GFP-Cre-ER fusion protein delivered with a LoxP-flanked-STOP tdTomato cassette. This system is tamoxifen-inducible such that once the Cre-ER fusion protein is stable under hypoxia, it activates permanent tdTomato expression [71]. This system was tested in HR1299-MR-derived tumors and validated by comparing the tdTomato expression with EF5 staining. 

Overall, fluorescent reporters are valuable methods for long-term monitoring of hypoxia in animal models of cancer. Moreover, this approach can be optimized to respond to a range of O_2_ concentrations by incorporating more or less O_2_-dependent elements (e.g., HRE, ODD, etc.). Like immunolabeling of HIF-related proteins or nitroimidazole probes, cells expressing a fluorescent reporter can be imaged at single-cell resolution [52]. However, while immunolabeling methods will only detect cells when they are experiencing hypoxia, fluorescent reporters can be imaged live over extended times using advanced intravital imaging or small animal fluorescent imaging. In addition, they can be engineered to mark hypoxic cells even when they become reoxygenated. The system’s downside is that distinguishing hypoxic from reoxygenated cells within the tumor typically requires the delivery of two distinct reporters.

### 3.4. Nitroreductase-Sensitive Fluorescent Probes

Fluorescent substrates activated by nitroreductase enzymes under hypoxia, as described in Section 2.5 can be delivered intravenously to murine models (4 in Figure 2). For instance, Zheng et al. developed a near-infrared (NIR) Cy-NO_2_ fluorescent substrate that nitroreductases can activate. They tested the Cy-NO_2_ by injecting it into mouse tumors derived from H22 cells. The NIR fluorescence could be imaged on a whole animal imaging device [72]. In another approach, Hettie et al. created a similar nitroreductase-activated NIR reporter, NO_2_-Rosol, which they tested in subcutaneous tumors derived from GBM39 glioblastoma cells. NO_2_-Rosol was delivered directly into the tumor, and fluorescence imaging was achieved with a CRi Maestro spectral fluorescent small animal imager. The study concluded that NO_2_-Rosol effectively reported intratumoral hypoxia, and its signal lasted for 90 min, which is a significantly higher retention time than other constructs utilized in this context [73]. Overall, NTR-sensitive substrates combined with fluorescent imaging can be used for real-time monitoring, and they have high sensitivity to changes in O_2_ levels [74]. 

### 3.5. Bioluminescence Imaging (BLI) Reporters

BLI measures light emitted by a living cell expressing a vector encoding a luciferase gene. In the presence of the substrate luciferin, luciferase causes the conversion of luciferin to oxyluciferin, resulting in light emission that a detector can collect. The amount of light detected is correlated with the level of hypoxia within the tumor [75] (5 in Figure 2). Saha et al. used this technology to generate a HIF-1α based reporter construct, 5HRE-ODD-Luc, which they transfected into MDA-MB-231 cells that were directly injected into the brain of female nude mice. After two weeks, D-luciferin was administered to each mouse, and images were acquired using a whole animal In Vivo Imaging System (IVIS). The study concluded that BLI could be detected in the hypoxic region of tumors [76]. Danhier et al. furthered this model by developing a PC3 cell line that constitutively expressed TdTomato. In addition, the cells had HIF-dependent expression of eGFP and a modified luciferase protein that contains an ODD. The bioluminescence signal was used to monitor acute hypoxia because the ODD domain caused rapid degradation of the luciferase enzyme upon reoxygenation. On the other hand, eGFP had a longer half-life and thus degraded more slowly upon reoxygenation, which allowed imaging of hypoxic cells even after reoxygenation. The combination of the bioluminescent and fluorescent reporters successfully allowed the distinction of hypoxic and post-hypoxic cells due to the difference in their half-life, which was reported to be 34 min and 15 h, respectively. However, contrary to BLI, the eGFP signal can be attenuated by variations in the tissue thickness during excitation and emission [18].

Whole animal fluorescent or BL imaging provides mm scale resolution images [77], but BLI requires the substrate luciferin to be injected immediately before imaging. The bioluminescent signal peaks between 7 and 20 min and then drastically wanes after injection [78]. Moreover, to obtain an accurate readout, luciferin must be well distributed within the tumor, which might be hindered, particularly in hypoxic regions, given the lack of functional vasculature in these areas. Finally, the equipment used for imaging is costly and is generally housed in a multi-user core facility rather than a single laboratory. 

### 3.6. Photoacoustic Imaging (PAI)

While fluorescent imaging can be hindered by low tissue penetration and light scattering, photoacoustic (PA) signals are generated by the absorption of near-infrared photons. PAI utilizes O_2_-sensitive dyes that initiate rapid thermoelastic expansion, which induces the propagation of sound waves recordable through a transducer to detect hypoxia via spectral absorbance differences [79] (6 in Figure 2). Shao et al. used the O_2_-sensitive dye methylene blue to test PAI in solid tissue using nude mice injected with LNCaP prostate cancer cells. When tumors reached 5 to 10 mm in diameter, the dye was injected directly into the tumor, producing images using ultrasound. The authors reported hypoxic gradients within the tumors confirmed with a needle-mounted O_2_ probe [80]. 

More recently, ratiometric PA probes have been designed, meaning the ratio between reacted and unreacted probes can be measured. For example, Knox et al. developed a reaction-based ratiometric PA probe with an N-oxide functionality that undergoes selective bioreduction by heme proteins under hypoxic conditions. This irreversible chemical reaction shifts the probe’s absorbance maximum and enhances the PA signal to improve detection. To test this approach, the authors implanted 4T1 cells subcutaneously into BALB/c mice to develop 300–400 mm^3^ tumors, and they were able to detect hypoxic gradients at high resolution and centimeter depths [81]. 

Gold nanorods have also been used as PA probes (AuNRs) since they have a strong absorbance in the NIR spectrum and good biocompatibility. Umehara et al. used AuNRs that contain nitroimidazole units on their surface to identify areas of hypoxia in xenograft tumors [82]. In summary, PAI is minimally invasive, which makes it promising for future clinical use. Although it allows the identification of hypoxia at a greater tissue depth than fluorescent reporters and other optical imaging modalities, deep tissue imaging using PAI is still a challenge. To date, this technology has been used to detect human breast tumors by virtue of their hypoxic status, suggesting that PAI O_2_ probes have the potential to be translated to the clinic and provide O_2_ readouts [83]. A tradeoff of this approach is the requirement for specialized probes that have to be injected into the subject.

### 3.7. Cherenkov-Excited Luminescence Imaging (CELI)

As described in Section 2.3, phosphorescent molecules such as Pt(II) and Pd(II) porphyrins and Ru(II) and Ir(III) complexes can provide dynamic monitoring of hypoxia via O_2_-quenching. CELI is an emerging imaging technology that measures PO_2_ via visible photons released during radiotherapy in conjunction with a phosphorescent probe (7 in Figure 2). One of the most well-established probes for CELI imaging is the O_2_-quenched dendritic molecule, PtG4. Cherenkov light generated from tissue subjected to radiation excites the phosphorescence of PtG4, and the phosphorescence decay time is a direct indicator of tissue PO_2_ [84,85]. Cao et al. developed an implantable agarose gel probe containing PtG4, which was directly injected into subcutaneous MDA-MB-231 cell-derived tumors. Tumors were imaged during fractioned radiotherapy delivered in four daily, 5 Gy doses. The retention time of PtG4 in the agarose gel was sufficient to measure pO_2_ at each treatment time [84]. 

Overall, there are still some obstacles to overcome before CELI can be used clinically to map intratumoral hypoxia, namely the toxicity associated with O_2_ probes. However, implantable PtG4 gel-probes delivered at an FDA-approved dose prescribed for mice was sufficient to image hypoxia in murine tumors [84]. Therefore, the most likely use for this technology would be to incorporate it into radiotherapy protocols that some cancer patients receive as a therapy to monitor intratumoral hypoxia as a way to monitor treatment efficacy.

### 3.8. Magnetic Resonance Imaging (MRI)

MRI is a noninvasive tool first employed to evaluate hypoxia in vivo in the later 20th century [86,87,88]. MRI uses a combination of magnets oriented in different directions to create a magnetic field that excites protons (i.e., hydrogen nuclei). The energy released during proton relaxation is captured and can be reconstructed into 3D images with a spatial resolution of 4–100 μm [89] (8 in Figure 2). Blood-oxygen-level dependent (BOLD) functional MRI measurements rely on regional differences in blood flow [90]. Oxyhemoglobin and deoxyhemoglobin have different paramagnetic properties. Deoxygenated hemoglobin is paramagnetic, which causes magnetic distortions in the surrounding tissue, whereas oxygenated hemoglobin is not. Therefore, image contrast in BOLD is determined by the local concentration of deoxy to oxyhemoglobin. BOLD fMRI has been successfully utilized to map hypoxia in animal models of cancer [91]. Still, the measurement can be confounded by several biological factors, such as heterogeneous tumor tissue, low regulation of blood flow, and variations in blood vessel size [92,93]. 

Tumor Oxygenation Level-Dependent (TOLD) MRI is a second MRI method that has been used to investigate hypoxia by measuring the concentration of free O_2_ molecules in the tissue [90,94]. O’Conner et al. recently published a review article detailing the differences between BOLD and TOLD MRI methodologies that extends beyond the scope of the current review [90]. Recent studies have incorporated siloxane injections in conjunction with TOLD MRI techniques to create precise measures of PO_2_ in animal models, called Proton Imaging of Siloxanes to map Tissue Oxygen Levels (PISTOL). PISTOL was developed by Kodibagkar et al. [95] and tested in vivo to measure O_2_ in the rat thigh muscle to develop higher resolution and faster MRI readouts [96]. To increase resolution, magnetic resonance contrast amplification (MR-CA) nanoprobes have also been utilized in MRI scans to improve the detection of hypoxia in xenograft models. The nanoprobes have been designed to respond to tumor acidosis, a common result of hypoxia, by triggering the release of incorporated contrast agents. Tumors derived from 4T1 breast cancer cells [97] and BxPC3 pancreatic cells [98] have been used to confirm the efficacy of MR-CA nanoprobes at detecting hypoxic regions. Although these methods require access to advanced and costly equipment, intratumoral hypoxia has been successfully detected and visualized in 3D. Moreover, the noninvasive nature of MRI makes it extremely attractive for preclinical and clinical uses.

### 3.9. Electron Paramagnetic Resonance Imaging (EPRI)

EPRI is an imaging system similar to MRI. However, while MRI maps the distribution of protons, EPRI measures unpaired electron spins of diffusible O_2_ using an injected spin probe to measure relaxation directly (9 in Figure 2). Thus, the energy released when the two unpaired electrons of molecular O_2_ collide with the probe’s unpaired electron is linearly proportional to the O_2_ concentration, allowing direct PO_2_ measurements [99]. EPRI probes last in the site for several months after injection, and tolerate serial imaging (over several hours), making it useful for identifying hypoxic regions in a live animal [100,101]. Furthermore, EPRI offers a sub-millimeter resolution of PO_2_, ensuring values can be compared between various regions of the tumor [102]. The applicability of EPRI has been evaluated using both orthotopic and transgenic murine models of breast cancer [103]. EPRI and the probe-tracer OX063 have been used to determine the effectiveness of evofosfamide to improve tumor oxygenation [101]. Moreover, the feasibility of EPRI for the detection of hypoxia was successfully assessed in mouse models of glioblastoma [104] and colon adenocarcinomas [105]. Overall, EPRI is accurate for tissue O_2_ imaging but limited by penetration depth (10 mm) and the requirement for tracer injections [106]. EPRI is an emerging technology that the FDA has not yet approved, showing promise in preclinical research.

### 3.10. Positron Emission Tomography (PET) 

PET is an imaging technology that, when used in conjunction with 2-nitroimidazole radiolabeling tracers and computerized tomography (CT), can indirectly detect hypoxia non-invasively in live animals (10 in Figure 2). PET agents are radioactive compounds quantitatively measurable through their beta-decay, which is converted into a 3D image. Intratumoral hypoxia has been detected via PET by utilizing nitroimidazole isotopes, such as ^18^F-fluoromisonidazole ^18^F-FMISO and ^18^F-EF5, which irreversibly bind to thiol groups on metabolic proteins at rates inversely proportional to O_2_ concentrations [107]. Hirata et al. recently reported that ^18^F-FMISO provides valuable prognostic data on survival and treatment response for patients with glioma and information on necrosis, vascularization, and permeability of the tumor [108]. Whereas ^18^F-FMISO is the traditional hypoxia PET tracer, other nitroimidazole radiotracers such as ^18^F-fluoroazomycin arabinoside (^18^F-FAZA) and ^18^F-HX4 have been preclinically and clinically tested. For instance, Peeters et al. found that ^18^F-FMISO and ^18^F-HX4 showed high spatial reproducibility, while ^18^F-FAZA and ^18^F-HX4 displayed higher sensitivity to acute hypoxia in rat tumors [109]. Likewise, ^18^F-FAZA has been reported as a more promising tracer than ^18^F-FMISO because of its improved biodistribution and enhanced tumor-to-background ratio [110]. Furthermore, clinical studies in patients with head and neck cancers have also demonstrated that ^18^F-HX4 has higher sensitivity and specificity, faster clearance, and shorter injection-acquisition time than ^18^F-FMISO [111]. Therefore, while ^18^F-FMISO was once the most viable option, new technological capabilities have led to multiple PET tracers. 

Current research also explores the possibility for PET tracers to bind to hypoxia-induced proteins instead of thiol groups. For instance, More et al. developed an ^18^F-PET radiotracer based on the carbonic anhydrase inhibitor drug acetazolamide. The authors tested this approach in 4T1 and HT-29 BALB/c skin xenograft models. Unfortunately, despite its theoretical feasibility, there was little tracer uptake in the tumors [112]. 

One significant impediment to PET is that tracers are not solely affected by hypoxic conditions. ^18^F-FMISO may have variable readouts due to biomolecular factors such as glycolytic byproducts [113]. ^18^F-FMISO accumulation varies with the abundance of glutathione S-transferase P1. Moreover, the polarization of tumor-associated macrophages, present in 50% of the tumor mass, can also affect uptake. M2 and M1 macrophages have higher and lower uptake of ^18^F-FMISO, respectively, compared to non-polarized M0 macrophages [114]. These conclusions were confirmed by O’Neill et al. who reviewed data on gross, molecular, and biological features across various cancer types [115]. A further limitation is the spatial resolution intrinsic to PET—while hypoxia can occur on the micron scale, PET is limited to 1–2 mm resolution [89]. Despite some limitations, PET is a well-established method that has been successfully utilized in the clinic to measure intratumoral hypoxia in multiple solid tumor types [116]. 

### 3.11. Electrochemical Oxygen Sensors

The “Clark electrode” O_2_ sensor developed in 1956 [117] has been considered the gold standard for O_2_ detection. A voltage is applied to the platinum electrode, causing O_2_ to be reduced on the surface of the electrode, thereby producing a measurable current directly proportional to O_2_ concentration [118]. Recent efforts have thus been underway to miniaturize electrochemical sensors to expand their use (11 in Figure 2). Rivas et al. recently tested a wireless, implantable Clark-type electrochemical O_2_ sensor in rabbits implanted in the right femoral quadricep muscle. The sensors distinguished between induced hyperoxia and hypoxia states by altering the fraction of inspired oxygen (FiO_2_) that the rabbits could breathe [119]. Gray et al. used a similar Clark-like sensor placed on the serosal surface of rats to study changes in O_2_ concentrations and visceral tissue O_2_ tension during intestinal surgery. The authors reported that the miniaturized sensor was better suited to obtain accurate readings than manual needle-electrodes, which can compress vasculature in a localized area and potentially affect the blood flow. Moreover, larger areas can be measured using multiple miniaturized sensors instead of a single large Clark electrode [120]. Overall, these sensors provide an accurate option to directly detect hypoxia, but applicability in the clinic is still exclusively limited to needle-probes. 

### 3.12. Invasive Optical Oxygen Sensors

Another needle-based method of measuring hypoxia in solid tumors is the use of invasive optical O_2_ sensors. These sensors contain an optical fiber coated with an O_2_-specific phosphorescent dye at the tip, as described in Section 2.6 (12 in Figure 2). Invasive optical sensors are frequently used as a precise method of confirming local O_2_ concentrations due to their high spatial resolution of 300 μm and 0.1 mmHg [121]. For example, fiber-optic needle microsensors have been utilized to measure PO_2_ levels in sarcoma [63] and pancreatic [122] murine models. Our studies also used a fixed-needle microprobe to measure O_2_ concentration as a function of penetration depth in an orthotopic tumor-derived with MDA-MB-231 cells [19]. 

These phosphorescent probes use weak excitation light to avoid the formation of the O_2_ singlet, which can be toxic to the surrounding tissue [123]. Moreover, external light, poor blood circulation, and pulse rate changes can be limiting factors. However, despite some potential for improvement, invasive optical O_2_ sensors have been successfully used for analyzing O_2_ gradients across tumors due to their precision and accuracy.

## 4. Discussion

Intratumoral hypoxia has a frequent incidence in solid primary tumors and greatly contributes to tumorigenesis and metastasis. In the early 1990s, multiple studies investigated the O_2_ levels in solid tumors utilizing polarographic probes, such as Clark probes, that are now commercially available from companies such as Eppendorf. They established that intratumoral hypoxia is an adverse indicator for patients with cancer [5]. Since then, in vitro models that progressed from 2D monolayer cell culture to 3D cellular clusters have been used to model physiological O_2_ and nutrient gradients found in vivo. In vivo models faithfully reproduce the lack of vasculature and subsequent hypoxic regions in solid tumors. Although mouse models are a standard in cancer research, larger animals, such as rats [124] or rabbits [125], better resemble human scale and physiology and have been used to test O_2_ detection methods. Furthermore, non-human primate models have been employed to assess biosafety parameters such as the biodistribution of radioactive hypoxia tracers for PET imaging [126] or to determine the effect of BOLD signal on neural activity [127]. As the relevance of intratumoral hypoxia increased, methods to assess O_2_ levels both in vitro and in vivo became critical. 

The selection of adequate methods to monitor O_2_ levels in experimental set-ups must take several factors into account, namely whether they can be performed in live animals or cells versus fixed tissues and the scale and sensitivity of the measurement. To facilitate comparison across the different methods described in this review, we categorized the techniques according to a series of features that we consider relevant and their applicability in different experimental settings (Table 1). A method can be selected to fit the scientific question or the available resources.

While O_2_-electrode or optical probes provide an accurate readout in situ, alternative methods have been established to overcome the need for an invasive approach to assess deeper tumors and preserve tissue integrity. Immunolabeling of hypoxia-regulated proteins in tissue sections has been extensively optimized to be performed in vitro and has also been employed in the clinic by staining human pathological specimens [134,135]. This particular approach is limited by rapid protein turnover and potential cross-regulation of these target proteins via mechanisms other than hypoxia. Another multifaceted approach requires the delivery of 2-nitroimidazole probes such as pimonidazole, which can be detected exclusively in hypoxic cells by immunolabeling. This method can be applied to basic cell culture or animal and human tissue sections [136]. However, immunolabeling techniques can only allow imaging of hypoxic regions when the tissue is resected and processed. Moreover, the staining is usually performed in a small portion of harvested and fixed tissue, which might not represent the entire tumor, particularly in the case of a human biopsy. 

Phosphorescent and nitroreductase-regulated fluorescent probes are an easy-to-implement alternative to investigate dynamic changes in hypoxia. These probes can be added to cell culture medium or delivered to the animal intravenously to be immediately measured. This feature makes them readily marketable, and several options are available commercially. Basic scientists have taken advantage of transcriptionally regulated fluorescent, and luminescent reporters to monitor the dynamics of hypoxic and reoxygenated cancer cells. These approaches require extensive design and optimization, which can be time-consuming. However, fluorescent markers have allowed us to isolate hypoxic and reoxygenated cells to begin to unravel their contribution to tumor progression and metastatic spread.

Recent advances in medical imaging have optimized imaging modalities such as MRI and PET to specifically detect O_2_ levels in a tumor. Basic scientists have kept up with these advances by utilizing these methods in preclinical models. Furthermore, in murine models, whole-body imaging has been extensively employed by utilizing BLI and PAI to enable time course studies in live animals. Although whole-body imaging allows dynamic monitoring of intratumoral hypoxia throughout long time-course experiments and better resembles clinical methods, the low spatial resolution prevents single-cell tracing. 

Recently, promising efforts have been made to improve O_2_ resolution and measure actual PO_2_ levels using PET. Furthermore, both CELI and EPRI are high O_2_-resolution (PO_2_) emerging technologies with clinical potential that are currently being optimized to obtain FDA approval. In summary, whole-body imaging methods allow dynamic monitoring of O_2_ levels in mice and are the most promising path for clinical use, but lower spatial and oxygen resolution or access to specialized high-cost equipment are limiting factors in preclinical research. Currently, PET offers the safest and cost-efficient option to be implemented in the clinic.

Hypoxia is a condition studied across multiple research fields, particularly in the context of developmental biology with focus on embryogenesis [137]. Aside from cancer, hypoxia also plays pathophysiological roles in myocardial ischemia, metabolic diseases, chronic heart and kidney diseases, and in reproductive disorders [138]. Although significant efforts have been conducted to detect and monitor hypoxic O_2_ levels experimentally, it is still challenging. Many of the described methods are only a proxy for actual levels of O_2_, can be challenging to adapt, or require access to specialized equipment. However, current techniques have been critical in establishing findings that have impacted the field of cancer research, namely: (1) the prognostic value of intratumoral hypoxia; (2) the hypoxic-upregulation of cancer hallmarks such as metabolism, migration, and chemoresistance; and (3) the significant contribution of cells that experienced intratumoral hypoxia to metastasis. As the role of hypoxia in cancer progression and resistance becomes more defined, multiple therapeutic approaches are being investigated pre-clinically and in clinical trials [139,140]. Thus, there is an urgent need to standardize methods to detect intratumoral hypoxia to (1) aid treatment decision and (2) assess whether hypoxia-targeted therapies perform as designed. The ultimate goal of this review is to aid researchers in selecting a suitable method to detect hypoxia in their studies by providing a side-by-side comparison across the most well-established and most recently optimized methods. 

## Figures and Tables

**Figure 1 cells-11-00686-f001:**
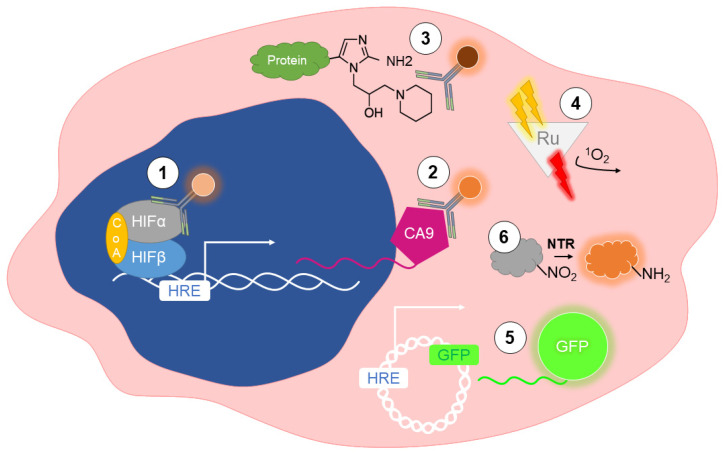
Detecting hypoxia in vitro. (**1**) Immunolabeling of HIF-1α or HIF-2α protein. (**2**) Immunolabeling of downstream transcriptional targets of HIFs (e.g., CA-IX). (**3**) Immunolabeling of hypoxia probes delivered exogenously 3 h before fixation (e.g., pimonidazole). (**4**) Oxygen quenches phosphorescent probes (e.g., Rubidium) after excitation with 2-photon light. (**5**) DNA construct is transcriptionally regulated by HIFs that express any protein (e.g., GFP) in a hypoxia-dependent manner. (**6**) Fluorescent molecules activated by nitroreductases (NTR) exclusively under hypoxia.

**Figure 2 cells-11-00686-f002:**
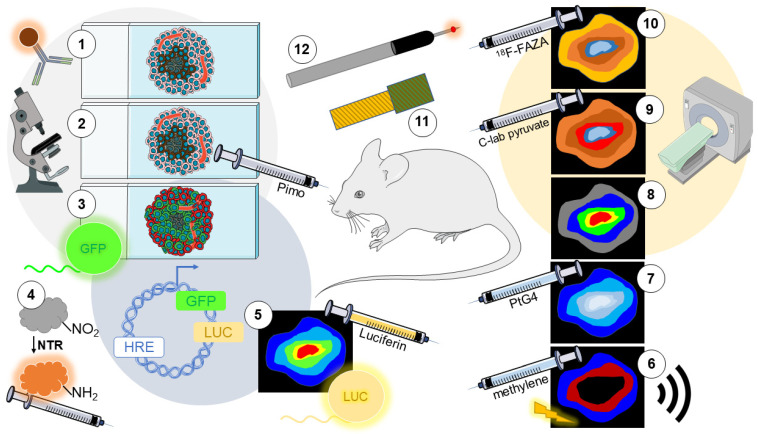
The detection of hypoxia in vivo. (**1**) Immunolabeling of endogenous hypoxia markers in tumor sections by IF/IHC. (**2**) Immunolabeling of exogenous hypoxia probes delivered via I.P./I.V. injection 1–2 h before sacrificing the animal (e.g., pimonidazole) in tumor sections by IF/IHC or FC of tumor-dissociated cell suspension. (**3**) Hypoxia-dependent fluorescent reports are expressed in cancer cells used to generate tumors. Fluorescence can be acquired using a whole animal imaging approach, tumors sections can be imaged by fluorescent microscopy, or tumor cell suspension analyzed by FC. (**4**) Delivery of fluorescent molecules activated by NTRs exclusively under hypoxia. (**5**) Hypoxia-dependent bioluminescent reporter expression in cancer cells used to generate tumors. An animal is pre-injected with luciferin and imaged using IVIS. (**6**) Photoacoustic signals generated by the absorption of near-infrared photons in chromophores of O_2_-sensitive dyes delivered to the animal cause thermoelastic expansion recordable through a transducer. (**7**) Imaging of an O_2_-quenched phosphorescent molecule injected into the animal (e.g., PtG4) with Cherenkov-Excited Luminescence Imaging (CELI) that measures visible photons during radiotherapy. (**8**) Functional Magnetic Resonance Imaging (fMRI) technology that measures metabolic function via variations in oxyhemoglobin and deoxyhemoglobin ratios using blood-oxygen-level-dependent (BOLD) or tumor oxygenation level-dependent (TOLD) contrast methods. (**9**) Electron Paramagnetic Resonance Imaging (EPRI) is similar to MRI, but it uses an injected spin probe (e.g., C-labeled pyruvate). (**10**) Positron Emission Tomography (PET) is an imaging technology that uses 2-nitroimidazole radiolabeling tracers (e.g., ^18^F-FAZA) with computerized tomography. (**11**) Electrochemical oxygen sensor that can be implanted or inserted into a needle probe and detects ionization of O_2_ atoms via a reduction reaction at an electrode. (**12**) Optical invasive sensors contain an optical fiber with an O_2_-specific phosphorescent dye coated on the tip that, when excited, the emitted light is captured optically, and the ratio of captured over emitted light is converted to a specific O_2_ value.

**Table 1 cells-11-00686-t001:** Compilation of methods to detect hypoxia. Partially adapted from [79,128]. Y = Yes; N = No; NA = Not Applicable; +/− = positive-negative readout. LM = Light Microscopy; FM = Fluorescent Microscopy; FC = Flow Cytometry; Temp = Temporal; Res = Resolution; Non-Inv = non-invasive.

Method	Detection	Live	Direct	Readout	Scale	Single Cell Res	Non -Inv	Dyna-mic	Temp. Res	In Vitro	Animal	Human	Processing
Endogenous markers	LM, FM, FC	N	N	+/−	μm [26]	Y	NA	N	NA	Y	Y	Y	Fixation Staining
Exogenous markers	LM, FM, FC	N	N	+/−	μm [66]	Y	NA	N	NA	Y	Y	Y	Fixation Staining
Fluorescent Reporter	FM, FC, Fluorescent imager	Y	N	+/− [79]	μm [79]	Y	Y	Y	ms [79]	Y	Y	N	Fixation Dissociation
NTR-sensitive Fluorescence	FM, FC, Fluorescent imager	Y	N	+/− [79]	μm [73]	Y	Y	Y	ms [79]	Y	Y	N	Fixation Dissociation
Phosphorescence CELI	FM, Fluorescent imager	Y	Y	pO_2_ [79]	μm [79]	N	Y	Y	s [128]	Y	Y	N	Pre-exposure
PAI	Ultrasound	Y	N	sO_2_ [79]	μm [79,129]	N	Y	Y	ms [79]	Y	Y	N	Pre-injection
BLI	Luminescent imager	Y	N	Intensity Gradient [130]	mm [77]	N	Y	Y	min [129]	Y	Y	N	Pre-injection
MRI	MRI machine	Y	N	B: deoxyHb [90] T: [O_2(s)_] [90]	mm [131]	N	Y	Y	s-min [128]	N	Y	Y	Pre-injection
EPRI	EPRI machine Spin tracers	Y	Y	pO_2_ [99]	mm [129,131]	N	Y	Y	min-hr [128,129]	N	Y	N	Pre-injection
PET	Radiolabeled Tracers	Y	N	radiotracer [129]	mm [131,132]	N	Y	Y	min-hr [133]	N	Y	Y	Pre-injection
Clark Electrode	Current meter	Y	Y	pO_2_ [121]	μm [121]	N	N	Y	s [121]	N	Y	Y	Implant/ Insertion
Invasive optical probes	Optical Detector	Y	Y	pO_2_ [121]	μm [121]	N	N	Y	ms [121]	Y	Y	Y	Insertion

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
