# Peer review of "Detection of Hypoxia in Cancer Models: Significance, Challenges, and Advances"

_cells, 2022, doi:10.3390/cells11040686_

Round 1

Reviewer 1 Report

This review nicely summarize an important topic related to hypoxia and cancer. It comes from a leading lab in the field, and presents an update view on this topic. 

The review is well-organized, nicely written, and overall is of high quality.

The only suggestions I can offer are the following:

  • It is not obvious from the manuscript's title that the focus is on cancer. It might be more accurate to specify that in the title.
  • I'm missing a bit more depth in regards to the molecular mechanisms on the cellular and molecular levels related to hypoxia. 
  • Hif1 undergoes post-translational modifications, and as such the transcript and the protein levels do not always behave the same on the molecular level. This should be emphasized, and examples related to the PTM should be cited. 
  • Perhaps brief description of hypoxia in other medical fields will be beneficial, to expand the readers' view to non-cancer hypoxia-related fields such as neurodevelopmental disorders, neuroscience-related fields, etc. 
  • Can the authors discuss hypoxia in animal models such as non-human primates: what is known, what will we gain from such study, etc.?
  • HBOT should be at least mentioned in a sentence, as a relevant potential treatment related to hypoxia (although not necessarily related to cancer). 
  • Also, a discussion on the gap in knowledge we have in the field of hypoxia and cancer, and how should it be studied, will improve the discussion and the manuscript's impact in the field. 

Author Response

This review nicely summarizes an important topic related to hypoxia and cancer. It comes from a leading lab in the field, and presents an update view on this topic. 

The review is well-organized, nicely written, and overall is of high quality.

We thank the reviewer for careful consideration of our work and for the helpful comments.

The only suggestions I can offer are the following:

  • It is not obvious from the manuscript's title that the focus is on cancer. It might be more accurate to specify that in the title.

The title has been modified to “Detection of hypoxia in cancer models: significance, challenges, and advances” to make this clear.

  • I'm missing a bit more depth in regards to the molecular mechanisms on the cellular and molecular levels related to hypoxia. 

We wanted the focus to be on current measurements of hypoxia used in cancer, but we have added a few more sentences in the introduction (lines 39-46) to address this concern and included the details that you requested in your third comment (below).

  • Hif1 undergoes post-translational modifications, and as such the transcript and the protein levels do not always behave the same on the molecular level. This should be emphasized, and examples related to the PTM should be cited. 
  • Perhaps brief description of hypoxia in other medical fields will be beneficial, to expand the readers' view to non-cancer hypoxia-related fields such as neurodevelopmental disorders, neuroscience-related fields, etc. 

In the discussion we have now included that measuring/detecting hypoxia is not just important in cancer but also in other pathophysiological conditions? (lines 644-647).

  • Can the authors discuss hypoxia in animal models such as non-human primates: what is known, what will we gain from such study, etc.?

We added the following comment to the discussion.

"Although mouse models are the standard in cancer research, larger models, such as  rat [137] or rabbit [138], better resemble the human scale and physiology and have been used to test O2 detection methods. Furthermore, non-human primate models have been employed to assess biosafety parameters such as the biodistribution of radioactive hypoxia tracers for PET imaging [139] or to determine the effect of BOLD signal on neural activity [140].”

  • HBOT should be at least mentioned in a sentence, as a relevant potential treatment related to hypoxia (although not necessarily related to cancer). 

We avoided mentioning potential treatments that target hypoxia or hypoxic cells because there are so many that it is beyond the scope of this manuscript.

  • Also, a discussion on the gap in knowledge we have in the field of hypoxia and cancer, and how should it be studied, will improve the discussion and the manuscript's impact in the field. 

We have added additional information on the gaps in detecting hypoxia in the discussion (lines 655-659).

Reviewer 2 Report

In their review “Detection of hypoxia: significance, challenges, and advances”, the authors provide a detailed overview of the most important experimental approaches to detect hypoxia in cell cultures and preclinical cancer models, and they assess the advantages and disadvantages of both established and new methods. The authors categorize these methods according to different characteristics and their applicability in different experimental settings, which enables the reader to choose a suitable method according to the scientific question and the available resources. In view of the potential clinical importance of hypoxia in cancer progression, this review is of interest to a broad readership. It is well-founded and comprehensible.

There are only a few minor points:

Line 40: “HIF1/2 hetereodimers” is misleading, because hypoxia-inducible factors HIF-1 and HIF-2 are different heterodimers, each consisting of an alpha and a beta subunit.

Line 90: Reference 28 seems inadequate to me as reference for the Warburg effect and aerobic glycolysis in cancer cells. Warburg’s original work from 1928 can be found here: https://link.springer.com/article/10.1007/BF01504608, but pointing to a more recent review (e.g., PMID: 26778478) on the subject would be more helpful to the reader.

Line 103: A reference would be helpful for the statement “multiple mechanisms other than hypoxia can enhance HIF-1α and HIF-2α expression”.

Line 173: The maturation of the GFP chromophore is oxygen-dependent, which is why reference should be made in this context to the work on UnaG cited below.

Line 175: Rather than the correct folding of the beta barrel structure of GFP, the maturation of its central chromophore is oxygen dependent.

Line 214: “fluoresce” instead of “fluoresce” (and a few more typos throughout the text)

Line 228/9: “a specific O2”  - “concentration”, “partial pressure”, or “level” seems to be missing?

Line 265: The same abbreviations should be used throughout. Here: CA-IX instead of CAIX?

Line 266: Rubidium may be used as a probe, but is not strictly a dye.

Line 440/1: “Deoxygenated hemoglobin is paramagnetic, whereas deoxygenated hemoglobin is not.” Replace the second “deoxygenated” in this sentence by “oxygenated”. Unlike paramagnetic deoxygenated hemoglobin, oxyhemoglobin is diamagnetic.

Line 571 ff: strange line breaks in the heading of the table

Line 573: explanation of the abbreviation “diss.” is missing

Line 578: Is there a difference between Eppendorf and Clark probes, or is Eppendorf just a manufacturer of Clark electrodes?

A list of abbreviations would be helpful.

I wonder if it would be helpful to the reader if the manufacturers/vendors mentioned in this review (e.g., Line 157: Thermofisher, Line 242: Pyroscience, …) were given a superscript number and their websites included in the list of references.

In the list of references, the capitalization should be used uniformly.

ref. 19: delete “2019 101”

ref. 28: see comment to line 90

ref. 43: the journal title is “Chemistry”

ref. 48: journal, volume and page numbers are missing

ref. 55: journal title is not abbreviated, volume and page numbers are missing

ref. 67: only the first page is indicated instead of “411-421”

ref. 68: give the volume without month

ref. 71: missing page number/article ID

ref. 74: missing page number/article ID

ref. 78: "April" is wrong. Missing volume and page number

ref. 81: missing page number/article ID

ref. 82: “e2001549” instead of “2001549”

ref. 84: wrong doi: „doi: 10.1021/acschembio.8b00099” instead of “doi:10.1021/ACSCHEMBIO.8B00099/SUPPL_FILE/CB8B00099_SI_001.PDF”

ref. 87: missing page number/article ID

ref. 89: author names?

ref. 91: author names?

ref. 92: wrong doi: delete “/BIBTEX”

ref. 96: only first page indicated

ref. 97: author names?

ref. 103: delete “https://www.jstor.org/stable/pdf/41433206.pdf. Accessed January 13, 2022.”

ref. 107: delete “JoVE (“

ref. 108: only first page indicated

ref. 111: volume and page numbers missing

ref. 112: delete “(8)”, article ID is missing

ref. 117: delete “2019, Vol 8, Page 1487“

ref. 119: journal title is „Pharmaceuticals“; delete “2019, Vol 12, Page 16”

ref. 120: only the first page is indicated; delete “/pmc/articles/PMC4074502/. Accessed January 10, 2022.”; doi is missing

ref 125: delete „https://home.liebertpub.com/ars.“

ref. 126: delete “June”; missing page number/article ID

ref. 127: delete “Vol 42, Issue 4, pp 731-734.”

ref. 132: delete “2017, Vol 7, Page 48.”

ref. 138: only first page indicated

Author Response

In their review “Detection of hypoxia: significance, challenges, and advances”, the authors provide a detailed overview of the most important experimental approaches to detect hypoxia in cell cultures and preclinical cancer models, and they assess the advantages and disadvantages of both established and new methods. The authors categorize these methods according to different characteristics and their applicability in different experimental settings, which enables the reader to choose a suitable method according to the scientific question and the available resources. In view of the potential clinical importance of hypoxia in cancer progression, this review is of interest to a broad readership. It is well-founded and comprehensible.

We thank the reviewer for providing helpful comments and careful editing to improve our review article.

There are only a few minor points:

Line 40: “HIF1/2 hetereodimers” is misleading, because hypoxia-inducible factors HIF-1 and HIF-2 are different heterodimers, each consisting of an alpha and a beta subunit.

To clarify, we have modified the sentence to say “Both HIF-1 and HIF-2 heterodimers recognize…”.

Line 90: Reference 28 seems inadequate to me as reference for the Warburg effect and aerobic glycolysis in cancer cells. Warburg’s original work from 1928 can be found here: https://link.springer.com/article/10.1007/BF01504608, but pointing to a more recent review (e.g., PMID: 26778478) on the subject would be more helpful to the reader.

We have changed reference 28 to refer to PMID: 26778478, as it is more contemporary than Warburg’s original work.

Line 103: A reference would be helpful for the statement “multiple mechanisms other than hypoxia can enhance HIF-1α and HIF-2α expression”.

We have now included PMID: 29230384 as a reference to support this statement.

Line 173: The maturation of the GFP chromophore is oxygen-dependent, which is why reference should be made in this context to the work on UnaG cited below.

Line 175: Rather than the correct folding of the beta barrel structure of GFP, the maturation of its central chromophore is oxygen dependent.

We incorporated this detail in lines 173-175 (now line177). And at line: 312 added “does not require O2 for chromophore maturation “ in context with the UnaG.

Line 214: “fluoresce” instead of “fluoresce” (and a few more typos throughout the text)

We have corrected the spelling to “fluoresce” (now line 217).

Line 228/9: “a specific O2”  - “concentration”, “partial pressure”, or “level” seems to be missing?

We added the word “concentration” to lines 228-229 (now 233).

Line 265: The same abbreviations should be used throughout. Here: CA-IX instead of CAIX?

We modified the figure legend to CA-IX from CAIX for consistency and checked the remaining abbreviations to ensure consistency.

Line 266: Rubidium may be used as a probe, but is not strictly a dye.

Thank you for the clarification. This has been corrected in the figure legend.

Line 440/1: “Deoxygenated hemoglobin is paramagnetic, whereas deoxygenated hemoglobin is not.” Replace the second “deoxygenated” in this sentence by “oxygenated”. Unlike paramagnetic deoxygenated hemoglobin, oxyhemoglobin is diamagnetic.

We have corrected this sentence by replacing the second deoxygenated to oxygenated on lines 440-441 (line 446 now).

Line 571 ff: strange line breaks in the heading of the table

The line break has been removed.

Line 573: explanation of the abbreviation “diss.” is missing

We added the full spelling for this so that it is no longer an abbreviation.

Line 578: Is there a difference between Eppendorf and Clark probes, or is Eppendorf just a manufacturer of Clark electrodes?

Eppendorf manufactures Clark electrodes, but it also offers optical sensors. We made this clarification in the discussion section.

A list of abbreviations would be helpful.

A list of abbreviations and acronyms was added at the end of the manuscript.

I wonder if it would be helpful to the reader if the manufacturers/vendors mentioned in this review (e.g., Line 157: Thermofisher, Line 242: Pyroscience, …) were given a superscript number and their websites included in the list of references.

We  don’t want to give any indication that we are endorsing a product, so we did not to include.

Thank you for careful editing of the references. We have made all of the suggested changes that you provided below as well as many more.

ref. 19: delete “2019 101” done

ref. 28: see comment to line 90 done

ref. 43: the journal title is “Chemistry”, the name is “Analytical Chemistry”

ref. 48: journal, volume and page numbers are missing done

ref. 55: journal title is not abbreviated, volume and page numbers are missing done

ref. 67: only the first page is indicated instead of “411-421” done

ref. 68: give the volume without month done

ref. 71: missing page number/article ID done

ref. 74: missing page number/article ID done

ref. 78: "April" is wrong. Missing volume and page number done

ref. 81: missing page number/article ID done

ref. 82: “e2001549” instead of “2001549” done

ref. 84: wrong doi: „doi: 10.1021/acschembio.8b00099” instead of “doi:10.1021/ACSCHEMBIO.8B00099/SUPPL_FILE/CB8B00099_SI_001.PDF” done

ref. 87: missing page number/article ID done

ref. 89: author names? done

ref. 91: author names? done

ref. 92: wrong doi: delete “/BIBTEX” done

ref. 96: only first page indicated done

ref. 97: author names? done

ref. 103: delete https://www.jstor.org/stable/pdf/41433206.pdf. Accessed January 13, 2022. done

ref. 107: delete “JoVE (“ done

ref. 108: only first page indicated done

ref. 111: volume and page numbers missing done

ref. 112: delete “(8)”, article ID is missing done

ref. 117: delete “2019, Vol 8, Page 1487“ done

ref. 119: journal title is „Pharmaceuticals“; delete “2019, Vol 12, Page 16” done

ref. 120: only the first page is indicated; delete “/pmc/articles/PMC4074502/. Accessed January 10, 2022.”; doi is missing done

ref 125: delete „https://home.liebertpub.com/ars.“ done

ref. 126: delete “June”; missing page number/article ID done

ref. 127: delete “Vol 42, Issue 4, pp 731-734.” done

ref. 132: delete “2017, Vol 7, Page 48.” done

ref. 138: only first page indicated done